# Long-Term Outcomes of Tachycardia-Induced Cardiomyopathy Compared with Idiopathic Dilated Cardiomyopathy

**DOI:** 10.3390/jcm12041412

**Published:** 2023-02-10

**Authors:** Moshe Katz, Amit Meitus, Michael Arad, Anthony Aizer, Eyal Nof, Roy Beinart

**Affiliations:** 1Sheba Medical Center, Ramat Gan 5266202, Israel; 2Sackler School of Medicine, Tel-Aviv University, Tel Aviv 6997801, Israel; 3NYU Grossman School of Medicine, New York, NY 10016, USA

**Keywords:** arrhythmia, cardiomyopathy, atrial fibrillation, heart failure, premature ventricular beats

## Abstract

Background: data on the natural course and prognosis of tachycardia-induced cardiomyopathy (TICMP) and comparison with idiopathic dilated cardiomyopathies (IDCM) are scarce. Objective: To compare the clinical presentation, comorbidities, and long-term outcomes of TICMP patients with IDCM patients. Methods: a retrospective cohort study of patients hospitalized with new-onset TICMP or IDCM. The primary endpoint was a composite of death, myocardial infarction, thromboembolic events, assist device, heart transplantation, and ventricular tachycardia or fibrillation (VT/VF). The secondary endpoint was recurrent hospitalization due to heart failure (HF) exacerbation. Results: the cohort was comprised of 64 TICMP and 66 IDCM patients. The primary composite endpoint and all-cause mortality were similar between the groups during a median follow-up of ~6 years (36% versus 29%, *p* = 0.33 and 22% versus 15%, *p* = 0.15, respectively). Survival analysis showed no significant difference between TICMP and IDCM groups for the composite endpoint (*p* = 0.75), all-cause mortality (*p* = 0.65), and hospitalizations due to heart failure exacerbation. Nonetheless, the incidence of recurrent hospitalization was significantly higher in TICMP patients (incidence rate ratio 1.59; *p* = 0.009). Conclusions: patients with TICMP have similar long-term outcomes as those with IDCM. However, it portends a higher rate of HF readmissions, mostly due to arrhythmia recurrences.

## 1. Introduction

Tachycardia-induced cardiomyopathy (TICMP), also known as arrhythmia-induced cardiomyopathy (AIC), is a subtype of acquired dilated cardiomyopathy [1]. TICMP has been documented in different forms of arrhythmia, including supraventricular tachycardia (SVT), atrial tachycardia (AT), ventricular tachycardia (VT), and frequent premature ventricular beats (PVBs), but most frequently it has been recognized in patients with atrial fibrillation (AF) and atrial flutter (AFL) [2].

TICMP is often considered a relatively benign cause of heart failure, and reversible when appropriate treatments are given in a timely fashion [3,4]. Treatment strategies mainly focus on rhythm control, following cardioversion and medication or ablation procedure, though rate control may also be implemented [5,6].

Short-term outcomes post-treatment usually demonstrate improvement in the left ventricular ejection fraction (LVEF) and left ventricular dimensions [7]. Notably, several studies reported an increased risk of sudden cardiac death, even following improvement in LV function [8,9]. However, data on the long-term outcomes of TICMP patients are lacking [9,10]. Both TICMP and IDCM are characterized by ventricular dilatation and depressed LV function and should be considered after ruling out hypertension, valvular, congenital, or ischemic heart disease [11]. They differ in etiology, baseline echocardiographic parameters, and reversibility of left ventricular systolic function with treatment. Yet, the prognosis and outcomes of these entities are unclear.

Objective: to compare the clinical presentation, comorbidities, and long-term outcomes of TICMP patients with IDCM patients.

## 2. Methods

### 2.1. Study Population

The study is a retrospective cohort analysis, aimed at comparing the prognosis of TICMP (*n* = 64) with IDCM (*n* = 66) patients. All electronic medical records of patients admitted to the Sheba Medical Center for heart failure and cardiomyopathy between March 2007 and June 2017 were enrolled. The etiologies causing heart failure or cardiomyopathy were defined. This initial database included subjects with new-onset cardiomyopathy. All patients were ≥18 years old and with LVEF ≤ 50% at presentation. Patients were defined as IDCM in the absence of any of the following conditions: ischemia, uncontrolled hypertension (>160/100), severe valvular disease, congenital heart disease, toxic exposure (chemotherapy/alcohol consumption, etc.), metabolic etiologies (nutritional deficiencies, endocrinopathy, etc.), or tachyarrhythmia. Patients were defined as TICMP if presented with heart failure secondary to arrhythmia without any other apparent causes for cardiomyopathy, and also showed an improvement of at least 15% in LVEF after rhythm control or rate control [12] within 6 months.

Once the cohort was established, a thorough investigation of each patient record was performed using the Sheba Medical Center computerized medical records. Demographic and clinical data and diagnostic imaging studies were collected and analyzed. In addition, the death date was retrieved from governmental mortality records.

During the index hospitalization, patients received guidelines-based therapy for heart failure, including angiotensin-converting enzyme inhibitors or angiotensin receptor blockers, beta-blockers, aldosterone receptor antagonists, digitalis, and diuretics if needed. In addition, patients with TICMP were treated with rhythm control or rate control strategies according to physicians’ discretion.

### 2.2. Study End Points

The primary endpoint was a composite of death, myocardial infarction, thromboembolic events, assist device, heart transplantation, and symptomatic VT/VF. The secondary endpoint was recurrent hospitalization due to heart failure exacerbation.

Study endpoints were evaluated at 5 years and for the whole length follow-up.

The Institutional Review Board of Sheba Medical Center approved this study.

### 2.3. Statistical Analysis

Comparisons between groups were analyzed by Chi-square or Fisher’s exact tests for categorical parameters, and Mann–Whitney test for continuous parameters. The non-normally distributed continuous variables were reported as a median and interquartile range [Q1–Q3]. Categorical variables were reported as numbers and percentages. The length of follow-up was described using the reverse censoring method. Kaplan–Meier curves were used to describe time to the primary endpoint, as well as recurrent hospitalization due to heart failure exacerbation. A log-rank test was used to compare survival between groups. Cox regression was done to estimate the crude hazard ratio for the primary and secondary endpoints. A propensity score was used to reduce the effects of confounding in the baseline characteristic between study groups. The propensity score for an individual is defined as the probability of being assigned to TICMP given all relevant covariates. The propensity score was calculated using logistic regression and then stratified into quintiles. The following variables were used to calculate the propensity score: age, sex, hypertension, smoking, cerebrovascular disease, diabetes mellitus, diabetes mellitus treated with insulin, peripheral vascular disease, valvular heart disease, ischemic heart disease, and LVEF.

To reduce the effect of confounders, stratified cox regression by the propensity score quintiles was performed for the primary and secondary endpoint. To compare the incidence of recurrent hospitalization during the follow-up period, Poisson regression was applied to estimate the incidence rate ratio. The natural logarithm of the length of follow-up was used as an offset variable in the Poisson model.

All statistical tests were two-tailed. *p* < 0.05 was considered statistically significant.

SPSS software was used for statistical analysis (IBM SPSS Statistics, Version 25, IBM Corp, Armonk, NY, USA, 2017).

## 3. Results

Sixty-four TICMP and 66 IDCM patients were hospitalized at the Sheba Medical Center with new onset heart failure between March 2007 and June 2017 (Figure 1). Baseline demographic characteristics and co-morbidities of the study population are shown in Table 1. In both groups, patients were predominantly males, overweight, and with a significant proportion of pre-existing hypertension. Patients in the TICMP group had a higher proportion of preexisting non-significant valvular heart disease (11% versus 2%, *p* = 0.03). There were no other significant differences in demographics and medical history. The most common etiology for TICMP was atrial fibrillation (76%), of whom 17 (27%) had a history of atrial fibrillation prior to admission (Appendix A).

Clinical characteristics: vital signs, NYHA function class, and echocardiographic and electrocardiographic findings, performed upon presentation, are shown in Table 2. Overall, patients in both groups presented with similar severity of symptoms, as evidenced by the NYHA function class. As expected, heart rates were much faster in TICMP patients, with a median heart rate of 120 bpm [IQR 90–132] versus 82 bpm [IQR 73–103] in idiopathic DCM patients (*p* < 0.001). There were no significant differences in laboratory study results. Prescription drug data showed higher use of beta-blockers prior to admission in the TICMP group compared with that of IDCM (55% versus 17%, *p* < 0.001). Anti-arrhythmic drug use was also higher in TICMP patients (20% versus 3%, *p* = 0.002), as well as statin use (41% versus 21%, *p* = 0.02).

Echocardiographic findings demonstrated, as expected, significant differences in left ventricle dimensions, with IDCM patients having larger dimensions. The left ventricular end-diastolic diameter in IDCM was 5.8 cm [5.5–6.4] versus 5.1 cm [4.6–5.6] in TICMP, *p* < 0.001, and left ventricular end-systolic diameter in IDCM was 5 cm [4.5–6.2] versus 3.8 cm [3.5–4.9] in TICMP, *p* < 0.001. Similarly, left ventricle mass was significantly higher in IDCM patients (231 g [206–268] versus 195 g [158–242], *p* < 0.001).

As for electrocardiographic findings, IDCM patients had a more frequent left bundle branch block pattern (32% versus 11%, *p* = 0.003). The QRS duration was also significantly longer compared with those of TICMP patients (106 ms [96–148] versus 98 ms [86–111], *p* = 0.002). Prolonged QRS duration (QRS duration equal to or greater than 110 milliseconds) was found in 26 IDCM patients versus 18 TICMP patients. However, the difference was not statistically significant. Interestingly, the PR interval was longer in TICMP patients, albeit within the normal range (174 ms [150–192] versus 158 ms [138–174], *p* = 0.003).

Overall coronary artery disease assessment, including invasive coronary angiography, coronary computed tomography angiography (CCTA), ergometry, and/or single photon emission computed tomography (SPECT) was performed in 64 (97%) IDCM patients and 53 (83%) TICMP patients, *p* = 0.007. Two patients (3%) in IDCM did not have an ischemic workup in our institution. One had an ischemic evaluation in a different institution and was lost to follow-up. His parameters were mainly used for baseline characteristics. The second had severe chronic kidney disease and refused ischemic evaluation.

During the index hospitalization, only 5 (8%) patients with IDCM presented with arrhythmia as a secondary finding rather than the cause for cardiomyopathy. For TICMP patients, a rhythm control strategy was chosen in 51 (80%) patients. The remaining were treated with rate control. Four of these were treated with pacemaker (PM) implantation as part of the pace and ablate strategy, two single-chamber PMs, and two cardiac resynchronization therapy pacemakers (CRTP).

Follow-up: data were typically obtained for all patients 3 to 6 months following discharge. The median LVEF during follow-up was much lower in IDCM compared with TICMP patients (35% [IQR 20–45%], 55% [IQR 47–60%], respectively, *p* < 0.001). LVEF improved during follow-up in both groups (Appendix A), however, TICMP patients had greater improvement with a median improvement of 25% [IQR 18–30%] compared with only 9% [IQR 5–15%] in IDCM (*p* < 0.001). During follow-up, more TICMP patients underwent pacemaker implantation (12 versus 1, *p* = 0.001): 10 out of 12 (83%) pacemakers were implanted as part of pace and ablate strategy (three CRTP, three dual-chamber PMs, and four single chamber PMs), and two dual chamber pacemakers for advanced atrioventricular block (AVB) and sick sinus syndrome. As expected, fewer TICMP patients were implanted with intracardiac defibrillators (5% versus 35%, *p* < 0.001) because the left ventricular ejection fraction recovered over time. Three TICMP patients underwent implantable cardioverter defibrillator (ICD) implantation during follow-up. One patient had very long QT with uncontrolled frequent premature ventricular beats and underwent dual chamber ICD implantation due to a high risk of Torsades de Pointes (TdP). The second patient underwent cardiac resynchronization therapy defibrillator (CRTD) implantation due to episodes of long short sequences leading to non-sustained TdP. The third patient underwent dual chamber ICD for secondary prevention after experiencing myocardial infarction and sustained ventricular tachycardia. Fourteen IDCM patients underwent CRTD implantation. Ten out of the fourteen patients who were implanted with CRTD had left bundle branch block (LBBB) at presentation.

### Study Endpoints

The primary composite endpoint and all-cause mortality were similar between TICMP and IDCM groups during a median follow-up time of 6.43 years [IQR 5.2–8.2]. During the follow-up, 24 of 130 (18%) patients died, 10 (15%) in the IDCM group and 14 (22%) in the TICMP group (Table 3). A Kaplan–Meier survival analysis showed no significant difference between TICMP and IDCM groups for event-free survival of the composite endpoint (log rank, *p* = 0.328) and all-cause mortality (Log Rank, *p* = 0.139) (Figure 2 and Figure 3). In univariate analysis, the mean time to readmission for heart failure exacerbation was shorter in the TICMP patients 5.2 years (95% CI 4–6.4) versus 6.9 years (95% CI 5.9–8) (log rank, *p* = 0.035) (Figure 4). However, this difference became statistically insignificant in multivariate analysis following propensity score adjustment (HR: 1.55; 95% CI 0.85–2.8; *p* = 0.15) (Table 4). Interestingly, Poisson regression analysis showed that the incidence of recurrent hospitalizations during the follow-up period was much higher in TICMP patients (incidence rate ratio 1.59, 95% CI 1.12–2.24; *p* = 0.009). The main trigger for heart failure exacerbation in TICMP patients was arrhythmia recurrence, while exacerbations in IDCM were mainly due to nonadherence to medical advice (Table 5). Nonadherence was defined as forgetting to take medications or skipping doses in the 2 weeks before arriving at the emergency department.

## 4. Discussion

In this study, we assessed the clinical presentation and prognosis of TICMP patients and compared them to IDCM patients. The main findings of the present study were: (1) All-cause mortality rates are similar between TICMP and IDCM; (2) The composite endpoint of death, myocardial infarction, thromboembolic events, assist device, heart transplantation, and VT/VF (primary endpoint) are similar between these groups; (3) Similar first readmission due to heart failure exacerbation; (4) Recurrent admission rates are significantly higher in TICMP patients mainly due to arrhythmia recurrences.

Our study represents a relatively large real-world cohort comprised of patients hospitalized with new onset cardiomyopathy (TICMP or IDCM). It adds information to previously published studies on TICMP. These studies, however, included relatively small sample size cohorts, and mainly from patients who underwent catheter ablations, possibly resulting in inherent selection bias [5,10].

Dilated cardiomyopathy belongs to the primary cardiomyopathies, disorders predominantly affecting the heart muscle, and is defined by the presence of left ventricular dilatation and systolic dysfunction in the absence of known abnormal loading conditions or significant coronary artery disease [13,14]. Genetic mutations can be found in up to 35% of dilated cardiomyopathy cases. Non-genetic causes such as drug toxicity, myocarditis, and more may result in similar clinical presentations [15,16]. When an underlying pathology cannot be identified, patients are diagnosed with IDCM [17].

The underlying cause of dilated cardiomyopathy determines the prognosis. Patients with IDCM usually have a better prognosis compared with patients with cardiomyopathy due to infiltrative disease, HIV infection, connective tissue disease, or doxorubicin [18].

TICMP is the result of prolonged and persistent tachycardia, and its prognosis is perceived by clinicians as relatively better. One expects that once the arrhythmia is controlled, recovery of LV function is seen over time. However, prolonged tachycardia results in elevated left ventricular filling pressures, impaired ventricular contractile function, reduced cardiac output, elevated systemic vascular resistance, and increased left ventricular wall stress. These hemodynamic changes lead to upregulation of the neurohormonal axis, which results in cellular and molecular changes. Furthermore, changes may remain even after improvement in the LV function and can serve as an arrhythmogenic substrate for arrhythmia recurrence [19]. In fact, animal models of tachycardia-induced cardiomyopathy demonstrated repolarization abnormalities, QT interval prolongation, polymorphic ventricular tachycardia, and sudden cardiac death. A human study supports these findings and implies a risk of sudden death even after controlling the heart rate and LVEF improvement [8].

In the present study, we found some major differences between IDCM and TICMP patients. These were mainly related to presenting symptoms, electrocardiography, and echocardiography measurements as well as some parameters during follow-up.

We found that the most common arrhythmia resulting in TICMP was atrial fibrillation (76%) (Appendix A). In approximately two-thirds of the patients, TICMP was the presenting symptom of atrial fibrillation, while others had a history of this arrhythmia. Not surprisingly, patients with TICMP had faster heart rates at presentation. Indeed, they were treated more with beta-blockers and antiarrhythmic drugs prior to their first admission. This suggests that TICMP can develop over time in patients with a recurrence of atrial fibrillation. Our findings are in keeping with previously published studies [8].

Electrocardiographic parameters were also different between the groups. Patients with IDCM had shorter PR intervals but wider QRS intervals with more frequent left bundle branch block patterns. This may lead by itself to cardiomyopathy or further deteriorate LV dysfunction regardless of the primary cause [20,21,22,23]. In this study, approximately 30% of patients with IDCM had LBBB, and half of them were treated with a bi-ventricular defibrillator (10 out of 21 patients). IDCM patients who received a bi-ventricular defibrillator had a small improvement in their ejection fraction after CRTD implantation. This fact strongly supports the diagnosis of IDCM over left bundle branch-mediated cardiomyopathy because in LBBB-mediated cardiomyopathy, the LVEF usually normalizes after CRTD implantation [23]. In addition, our data are in line with previous registry data that found that LBBB is common in patients with heart failure [24]. Vera et al. examined electrocardiogram and cardiac magnetic resonance (CMR) parameters of patients admitted for heart failure with reduced LVEF and concomitant supraventricular tachycardia [25]. Findings were analyzed to predict LVEF recovery. Like our cohort, they found that patients with dilated cardiomyopathy (DCM) had wider QRS than patients with TICMP. On CMR, the TICMP presented with higher LVEF whereas late gadolinium enhancement (LGE) was more frequent in dilated cardiomyopathy. QRS ≥ 100 ms, LVEF < 40% on CMR, and the presence of LGE were independent predictors of lack of LVEF recovery. In addition, during follow-up, DCM patients were more frequently admitted for heart failure than TICMP. In contrast to that, the LVEF in our cohort was not statistically different in both groups and we found that patients with TICMP were more frequently admitted for heart failure than DCM. The difference in heart failure re-admissions between studies can be explained by the different definitions of DCM and TICMP in the studies and the differences in presenting LVEF. In our cohort, we included only idiopathic dilated cardiomyopathy and not all patients diagnosed with dilated cardiomyopathy. Patients who have dilated cardiomyopathy secondary to other conditions may have a worse prognosis than patients with idiopathic dilated cardiomyopathy depending on the underlying etiology. Moreover, TICMP was defined as recovery to LVEF above 50% while our definition required an improvement of at least 15% in LVEF after rhythm control or rate control. Consequent to that, our TICMP cohort included patients with LVEF lower than 50% on follow-up (25% of TICMP patients had LVEF less than 47%). Lower LVEF improvement results in more heart failure hospitalizations [26]. The combination of heterogeneity, lower presenting LVEF in the DCM group, and including only TICMP patients with LVEF above 50% on follow-up contributed to the different results in these studies.

Dissimilarities in echocardiography measurements were also noted between the groups. Larger left ventricular dimensions and left ventricular mass were found in patients with IDCM. Jeong et al. reported that the initial echocardiographic parameters, especially the left ventricular end diastolic dimension can help differentiate TICMP from IDCM [12]. Similarly, we found larger left ventricular dimensions; however, there is a substantial overlap between the groups, and distinguishing between them solely based on ECG or echocardiography parameters could be challenging and at times misleading.

Importantly, with appropriate guideline-based medical therapy, including heart failure medications, treatment with antiarrhythmic drugs, or catheter ablation when appropriate, the LVEF increased in both groups. Notably, this improvement was substantially higher in the TICMP group during follow-up.

In terms of prognosis, no differences were found in respect of the primary and secondary endpoints. Hence, both mortality rates and the composite end point of death, myocardial infarction, thromboembolic events, assist device, heart transplantation, and VT/VF were similar between the groups. A possible explanation might be the higher proportion of AF in our cohort, an independent predictor of morbidity and mortality [27]. Hence, the “AF effect” occurring in TICMP patients might counterbalance the improvement in LVEF following appropriate treatment. Surprisingly, despite a high prevalence of AF in the TICMP group, we found similar rates of thromboembolic events in both groups. This could be attributed to both adherence to anticoagulation treatment in patients with AF and high risk for stroke, and a significant improvement in LVEF over time. In contrast, patients with IDCM were not routinely treated with anticoagulation, unless otherwise indicated, and had only slight improvement in LVEF, which can lead to venous stasis and subsequently to the creation of de novo mural thrombi [28]. Interestingly, the risk for first heart failure exacerbation was similar between both groups during the follow-up, although patients with TICMP had higher rates of recurrent HF-related hospitalizations during follow-up. The latter is mainly related to tachyarrhythmia recurrence (Table 5). Ahmad et al. reported a 50% recurrence rate of arrhythmia in patients with TICMP over a median follow-up of 6 months. They found that recurrence of arrhythmia was significantly associated with heart failure hospitalizations with an odds ratio of 6.65. However, after adjusting for other clinical characteristics, this association was not significant [29]. The lack of correlation between arrhythmia recurrence and heart failure hospitalizations in the multivariate analysis may stem from the way they defined arrhythmia recurrence and the clinical setting that it occurred in. Arrhythmia recurrence was based on a premature ventricular burden exceeding 10% or a 30-s episode of atrial fibrillation or atrial flutter during ambulatory monitoring. Therefore, they could have included patients with short-lived asymptomatic arrhythmia in the outpatient settings. In contrast, in our study, we examined the triggers for heart failure hospitalizations in TICMP patients in inpatient settings. Here, arrhythmia recurrence was the main cause of heart failure exacerbation and hospitalization.

Our finding suggests that the clinical prognosis of these two groups of patients is similar. Hence, TICMP should not be regarded by clinicians as a benign disorder. We believe that implementation of the guidelines’ recommendations of an early invasive strategy together with tight patient monitoring could lead to a reduction in clinical events and potentially improve prognosis in selected patients [30].

### Study Limitations

This retrospective study comparing long-term outcomes of TICMP and IDCM in patients hospitalized with new onset heart failure has several limitations. First, this study contains single-center data with a small number of patients enrolled. However, our sample size is relatively large in comparison with previous studies and only a few multicenter studies included a larger study population [31,32]. Second, the follow-up clinical data were retrieved from Sheba Medical Center records only. This could lead to a possible underestimation of clinical events that patients presented with them to a different hospital. However, the majority of patients admitted to Sheba Medical Center are nearby residents that would probably be readmitted to the same medical center when experiencing a recurrence of their heart condition. Moreover, the outpatient clinic follow-up data include information about recent hospitalizations in different hospitals. By reviewing the outpatient clinic follow-up, we were able to minimize this underestimation.

Third, in our cohort, we did not find a significant difference in mortality between groups, but a larger sample size and longer follow-ups are needed to validate our result. Fourth, data on all-cause mortality were retrieved from government records. These data do not include the etiology of death. Therefore, we could not rule out differences in cardiovascular mortality between groups.

Fifth, the cohort enrolled patients from 2007 to 2017 where a more conservative treatment strategy was taken, and an ablation catheter was not considered the treatment of choice. Implantation of recent guidelines from 2020 which advocates catheter ablation to reverse LV dysfunction when TICMP is highly probable might improve the outcome of TICMP and reduce heart failure hospitalization [30].

## 5. Conclusions

TICMP, though at first glance appears to be a relatively benign process, has similar long-term outcomes as IDCM even following the improvement of LVEF. In fact, it portends higher rates of worsening heart failure readmissions, mostly due to arrhythmia recurrences. Further studies are needed to evaluate whether early intervention and tight rhythm monitoring can lead to a better prognosis.

## Figures and Tables

**Figure 1 jcm-12-01412-f001:**
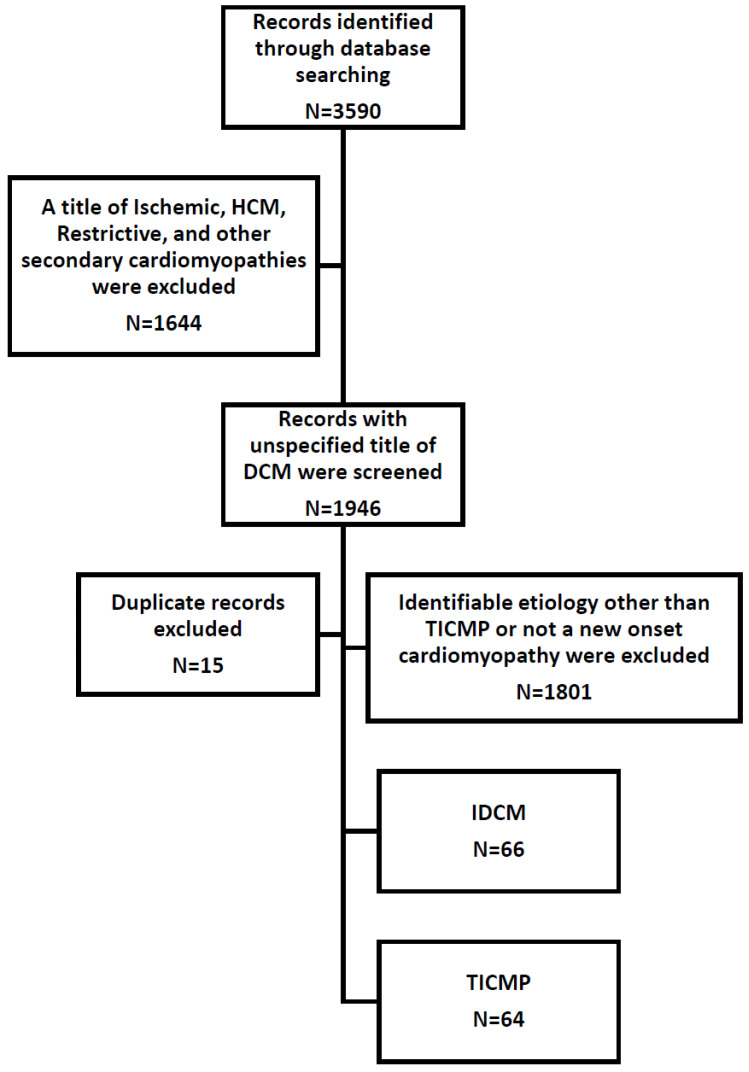
Flow chart of study population. DCM—dilated cardiomyopathy; HCM—hypertrophic cardiomyopathy; IDCM—idiopathic dilated cardiomyopathy; TICMP—tachycardia-induced cardiomyopathy.

**Figure 2 jcm-12-01412-f002:**
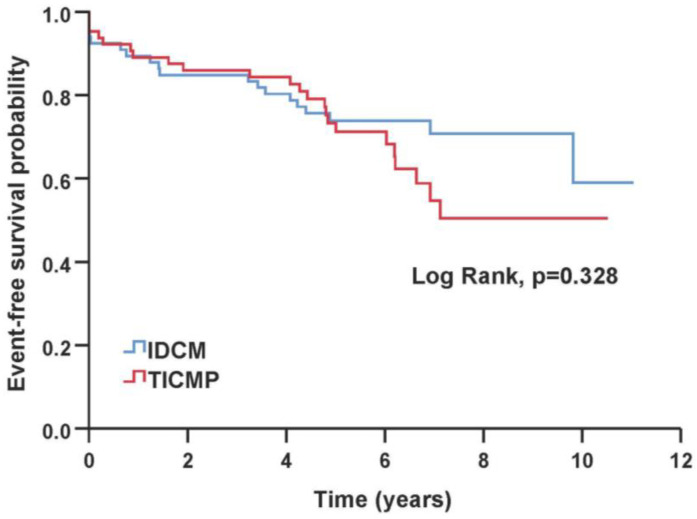
Kaplan–Meier curve of time to primary composite endpoint. IDCM—idiopathic dilated cardiomyopathy; TICMP—tachycardia-induced cardiomyopathy.

**Figure 3 jcm-12-01412-f003:**
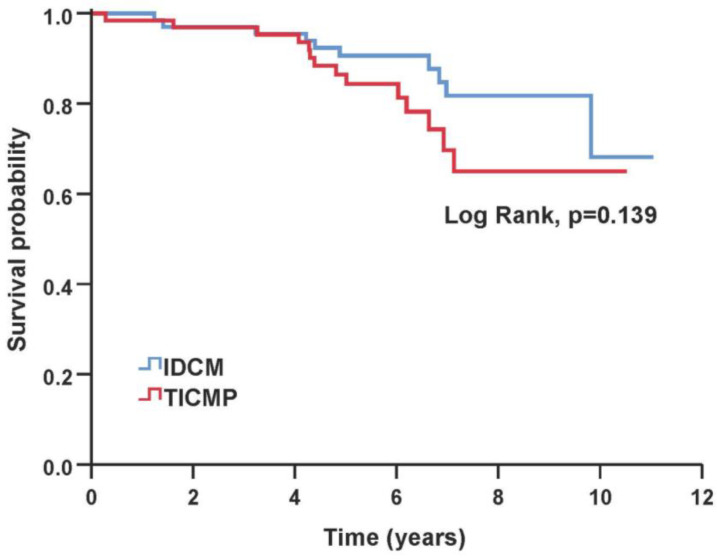
Kaplan–Meier estimates of overall survival based on the type of cardiomyopathy. IDCM—idiopathic dilated cardiomyopathy; TICMP—tachycardia-induced cardiomyopathy.

**Figure 4 jcm-12-01412-f004:**
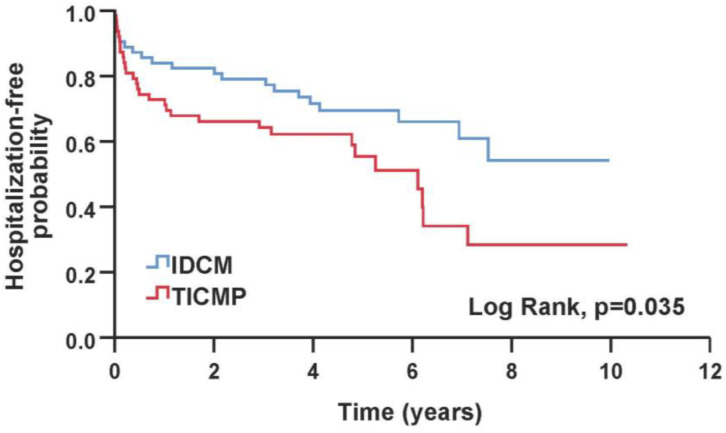
Kaplan–Meier survival probabilities for patients admitted for a first heart failure hospitalization. IDCM—idiopathic dilated cardiomyopathy; TICMP—tachycardia-induced cardiomyopathy.

**Table 1 jcm-12-01412-t001:** Baseline demographic characteristics and co-morbidities.

	IDCM *N* = 66	TICMP *N* = 64	*p*-Value
Sex, male; *n* (%)	51 (77)	41 (64)	0.1
Age, years; median [IQR]	60 [47–69]	65 [57–69]	0.09
BMI, kg/m^2^; median [IQR]	29 [24–33]	29 [26–33]	0.36
Hypertension, mmHg; *n* (%)	33 (50)	32 (50)	1
Current smoker; *n* (%)	15 (23)	7 (11)	0.08
CVD; *n* (%)	6 (9)	9 (14)	0.38
DM; *n* (%)	13 (20)	12 (19)	0.89
DM insulin Tx; *n* (%)	3 (5)	2 (3)	0.67
PVD; *n* (%)	6 (9)	3 (5)	0.49
VHD; *n* (%)	1 (2)	7 (11)	0.03 *
IHD; *n* (%)	6 (9)	5 (8)	0.79

BMI—body mass index; CVD—cerebrovascular disease; DM—diabetes mellitus; IDCM—idiopathic dilated cardiomyopathy; IHD—ischemic heart disease; IQR—interquartile range; PVD—peripheral vascular disease; TICMP—tachycardia-induced cardiomyopathy Tx-therapy; VHD—valvular heart disease. Chi-square or Fisher’s exact tests were performed for categorical parameters. Mann–Whitney test was performed for continuous parameters. Significant *p*-value was marked by *.

**Table 2 jcm-12-01412-t002:** Clinical characteristics at presentation.

	IDCM *N* = 66	TICMP *N* = 64	*p*-Value
Heart rate, BPM; median [IQR]	82 [73–103]	120 [90–132]	<0.001 *
SBP, mmHg; median [IQR]	132 [120–144]	127 [117–139]	0.16
DBP, mmHg; median [IQR]	75 [67–93]	83 [70–92]	0.44
NYHA FC; *n* (%)			0.34
I	2 (3)	2 (3)	
II	25 (39)	30 (47)	
III	26 (40)	27 (42)	
IV	12 (18)	5 (8)	
Laboratory results			
LDL, mg/dL; median [IQR]	105 [87–133]	105 [86–127]	0.7
HDL, mg/dL; median [IQR]	38 [32–46]	40 [32–47]	0.86
Triglycerides, mg/dL; median [IQR]	109 [77–150]	104 [74–147]	0.54
HGB, g/dL; median [IQR]	13.7 [12.5–14.8]	13.6 [12.5–14.6]	0.68
TSH, mIU/L; median [IQR]	1.8 [1.26–2.6]	1.95 [1.37–3.18]	0.47
Creatinine, mg/dL; median [IQR]	1.04 [0.88–1.25]	1.05 [0.89–1.26]	0.62
Albumin, g/dL; median [IQR]	3.9 [3.7–4.2]	3.8 [3.6–4]	0.06
Medical treatment			
Beta-blockers; *n* (%)	11 (17)	35 (55)	<0.001 *
CCBs; *n* (%)	5 (8)	5 (8)	0.96
Statins; *n* (%)	14 (21)	26 (41)	0.02 *
ACE inhibitors; *n* (%)	12 (18)	16 (25)	0.34
ARBs; *n* (%)	6 (9)	11 (17)	0.17
Digitalis; *n* (%)	0	1 (3)	0.24
AADs; *n* (%)	2 (3)	13 (20)	0.002 *
Echocardiography			
LVEF, %; median [IQR]	25 [15–35]	30 [20–32]	0.11
LVEDD, cm; median [IQR]	5.8 [5.5–6.4]	5.1 [4.6–5.6]	<0.001 *
LVESD, cm; median [IQR]	5 [4.5–6.2]	3.8 [3.5–4.9]	<0.001 *
LA diameter, cm; median [IQR]	4.6 [4.1–4.8]	4.5 [4.2–4.8]	0.98
LV mass, g; median [IQR]	231 [206–268]	195 [158–242]	<0.001 *
SPAP, mmHg; median [IQR]	42 [32–50]	40 [35–46]	0.8
Electrocardiography			
QRS duration, ms; median [IQR]	106 [96–148]	98 [86–111]	0.002 *
LBBB; *n* (%)	21 (32)	7 (11)	0.003 *
RBBB; *n* (%)	2 (3)	6 (10)	0.13
PR interval, ms; median [IQR]	158 [138–174]	174 [150–192]	0.003 *
QTc, ms; median [IQR]	475 [452–501]	467 [436–491]	0.17

AADs—antiarrhythmic drugs; ACE—angiotensin-converting enzyme; ARBs—angiotensin II receptor blockers; BPM—beats per minute; CCBs—calcium channel blockers; DBP—diastolic blood pressure; HDL—high-density lipoprotein; HGB—hemoglobin; IDCM—idiopathic dilated cardiomyopathy; LA—left atrium; LDL—low-density lipoprotein; LVEDD—left ventricular end-diastolic diameter; LVEF—left ventricular ejection fraction; LVESD—left ventricular end-systolic diameter; NYHA FC—New York Heart Association Functional Classification; IQR—interquartile range; SBP—systolic blood pressure; SPAP—systolic pulmonary artery pressure TICMP—tachycardia-induced cardiomyopathy; TSH—thyroid-stimulating hormone. Chi-square or Fisher’s exact tests were performed for categorical parameters. Mann–Whitney test was performed for continuous parameters. Significant *p*-value was marked by *.

**Table 3 jcm-12-01412-t003:** Composite outcome events.

	IDCM *N* = 66	TICMP *N* = 64	*p*-Value
All-cause mortality; *n* (%)	10 (15)	14 (22)	0.37
Acute coronary syndrome; *n* (%)	4 (6)	4 (6)	1
Thromboembolic events; *n* (%)	7 (11)	4 (7)	0.39
LVAD/heart transplant; *n* (%)	1 (2)	0	NA
Symptomatic VT/VF; *n* (%)	2 (3)	4 (6)	0.44

IDCM—idiopathic dilated cardiomyopathy; LVAD—left ventricular assist device; NA—not applicable; TICMP—tachycardia-induced cardiomyopathy; VT/VF—ventricular tachycardia or ventricular fibrillation. Fisher’s exact test was performed for categorical parameters.

**Table 4 jcm-12-01412-t004:** Crude and adjusted hazard ratios of the study endpoints.

Outcome	Crude		Propensity Score	
	HR 95% CI	*p*-Value	HR 95% CI	*p*-Value
Whole follow-up				
Composite endpoint	1.35 (0.74–2.4)	0.33	1.11 (0.57–2.18)	0.75
All-cause mortality	1.84 (0.81–4.17)	0.15	1.25 (0.49–3.17)	0.65
HF hospitalization	1.81 (1.03–3.18)	0.04 *	1.55 (0.85–2.8)	0.15
5 years follow-up				
Composite endpoint	0.97 (0.49–1.93)	0.94	0.84 (0.4–1.75)	0.64
All-cause mortality	1.47 (0.51–4.25)	0.47	1.09 (0.35–3.4)	0.88
HF hospitalization	1.6 (0.87–2.94)	0.13	1.43 (0.75–2.71)	0.28

CI—confidence interval; HF—heart failure. Cox regression was done to estimate the crude hazard ratio for the primary and secondary endpoints. Significant *p*-value was marked by *.

**Table 5 jcm-12-01412-t005:** Triggers for 1st heart failure recurrence.

	IDCM *n* = 21 *N* = 66	TICMP *n* = 30 *N* = 64
No trigger identified; *n* (%)	5 (8)	2 (3)
AF/AFL; *n* (%)	1 (2)	22 (34)
Bradyarrhythmia; *n* (%)	1 (2)	1 (2)
SVT; *n* (%)	0	1 (2)
Infection; *n* (%)	3 (5)	1 (2)
Concurrent lung disease; *n* (%)	1 (2)	0
Nonadherence; *n* (%)	7 (11)	1 (2)
Acute kidney injury; *n* (%)	1 (2)	0
Post-surgery volume overload; *n* (%)	1 (2)	1 (2)
Post-myocardial infarction; *n* (%)	1 (2)	0
Ventricular tachycardia; *n* (%)	0	1 (2)

AF—atrial fibrillation; AFL—atrial flutter; IDCM—idiopathic dilated cardiomyopathy; SVT—supraventricular tachycardia; TICMP—tachycardia-induced cardiomyopathy. Twenty-one IDCM patients and thirty TICMP patients had heart failure recurrence during follow-up. Percentages represent the absolute number of heart failure recurrences from the total cohort in each group.

## Data Availability

Data are available upon request.

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
