# Peer review of "Long-Term Outcomes of Tachycardia-Induced Cardiomyopathy Compared with Idiopathic Dilated Cardiomyopathy"

_jcm, 2023, doi:10.3390/jcm12041412_

Round 1

Reviewer 1 Report

I had a great pleasure of reviewing the manuscript entitled „Long-term outcomes of tachycardia induced cardiomyopathy compared with idiopathic dilated cardiomyopathy” submitted to the Journal of Clinical Medicine. The subject of tachycardia-induced cardiomyopathy is of special interest to anyone involved in the management of patients with heart failure, since the possible reversibility of LV dysfunction by cessation of tachycardia may bring spectacular benefits in patients even in a bad clinical condition.
The Authors present an elegant manuscript on the outcomes of patients with tachycardia-induced cardiomyopathy (TICM), compared with patients with idiopathic dilated cardiomyopathy. However, I have noted some areas which would require comment/modification:

-The authors define the criterion for DICM and TICM as not related to ischemia, while in the baseline characteristics section respectively 9% and 8% of patients have IHD. How would the authors explain that those patients are not classified as ischemic cardiomyopathy?

-I find the numbers of patients quite interesting - the prior reports demonstrate lower prevalence of patients with TICM in population of patients with non-ischemic HF. Here the numbers of patients with TICM and IDCM are almost identical, although the Authors state in the methods section that it were the consecutive patients enrolled.

-The criterion of reversibility of LVEF by >15% after anti-arrhythmic management is lacking the time threshold for its occurrence. Usually the maximal time assessed in prior studies with TICM was 6 months, but what was it in the submitted manuscript?

-The authors do not present the source of the endpoint data apart from mortality, which was based on the government mortality data. If the data came from Sheba Medical Center computerized medical records, could the authors discuss the possible underestimation of data for each specific endpoint, e.g. in case of presentation in a different hospital in Israel?

-How were VT/VF defined? This endpoint seems rather difficult to defend, since many VTs could be subclinical, and stable VT could be undetectable in the absence of devices with monitoring capabilities (no information provided on exact CIED implantation rates in patients from both cohorts, especially in DICM cohort in whom ICD is routinely recommended)?

-When were ECG recording analyzed? Was it an initial ECG on admission? If so, it should be noted that for instance, PQ interval is rate-dependent, and it is prolonged with higher rates. Could the authors also comment on that issue?

-Four patients from TICM cohort received pacemakers - what kind of devices were chosen in those patients for pace and ablate strategy? Were they CRT devices or conventional pacemakers? Moreover, in Discussion the data on the % of patients with LBBB and CRT devices in DICM are provided, which are not listed in any Table/Results section

-What were the indications for implantation of an ICD in patients with TICM? If the criterion of >15% LVEF increase was fulfilled were they the subjects with initially so decreased LVEF that even after recovery of LV the LVEF remained <35%?

-In Table 5 the data do not sum up to 100% in any of the columns. Could the authors explain this statistics? Was it that the sums should not include respectively 66 and 64, but only No of patients who experienced HF hospitalization?

-No cause of death is presented, which in my opinion should be considered a limitation, especially when analyzing the outcome-oriented study.

Author Response

Reviewer#1

Comments and Suggestions for Authors

I had a great pleasure of reviewing the manuscript entitled „Long-term outcomes of tachycardia induced cardiomyopathy compared with idiopathic dilated cardiomyopathy” submitted to the Journal of Clinical Medicine. The subject of tachycardia-induced cardiomyopathy is of special interest to anyone involved in the management of patients with heart failure, since the possible reversibility of LV dysfunction by cessation of tachycardia may bring spectacular benefits in patients even in a bad clinical condition.

The Authors present an elegant manuscript on the outcomes of patients with tachycardia-induced cardiomyopathy (TICM), compared with patients with idiopathic dilated cardiomyopathy. However, I have noted some areas which would require comment/modification:

-The authors define the criterion for DICM and TICM as not related to ischemia, while in the baseline characteristics section respectively 9% and 8% of patients have IHD. How would the authors explain that those patients are not classified as ischemic cardiomyopathy?

Our cohort included 3590 patients admitted from January 2007 to June 2017. We excluded patients with known cardiomyopathies and/or those with cardiomyopathies secondary to known etiologies (i.e. Myocardial infarction, HCM, restrictive cardiomyopathy or any secondary cardiomyopathy).  This analysis included 130 patients with newly diagnosed cardiomyopathy. Although these patients had history of CAD, their cardiomyopathy was not related to ischemia or myocardial infarction.

-I find the numbers of patients quite interesting - the prior reports demonstrate lower prevalence of patients with TICM in population of patients with non-ischemic HF. Here the numbers of patients with TICM and IDCM are almost identical, although the Authors state in the methods section that it were the consecutive patients enrolled.

The patients’ number were indeed similar, though we included only those who had newly diagnosed IDCM or TICMP. These numbers do not representative the incidence or prevalence of these etiology in the entire population.  

-The criterion of reversibility of LVEF by >15% after anti-arrhythmic management is lacking the time threshold for its occurrence. Usually the maximal time assessed in prior studies with TICM was 6 months, but what was it in the submitted manuscript?

Yes, as mentioned in result section (the follow-up subsection), the follow-up data was typically obtained for all patients 3 to 6 months following discharge. Meaning the reversibility of LVEF was re-assessed within 6 months. For clarification, this was added in the methods section (page 4 row 81).

-The authors do not present the source of the endpoint data apart from mortality, which was based on the government mortality data. If the data came from Sheba Medical Center computerized medical records, could the authors discuss the possible underestimation of data for each specific endpoint, e.g. in case of presentation in a different hospital in Israel?

The clinical end points were taken from the medical records at the Sheba Medical Center. Although the majority of patients admitted to Sheba medical are nearby residents and would probably be readmitted to the same medical center when experiencing these endpoints, we agree with the reviewer that some could be admitted to a different hospital. We add this statement to the limitation section (page 12 row 283-288)

-How were VT/VF defined? This endpoint seems rather difficult to defend, since many VTs could be subclinical, and stable VT could be undetectable in the absence of devices with monitoring capabilities (no information provided on exact CIED implantation rates in patients from both cohorts, especially in DICM cohort in whom ICD is routinely recommended)?

We thank the reviewer for his important comment. These were either unstable VT/VF or clinically relevant VT causing symptoms. We changed the manuscript accordingly to “symptomatic VT/VF” (page 4 row 93).

-When were ECG recording analyzed? Was it an initial ECG on admission? If so, it should be noted that for instance, PQ interval is rate-dependent, and it is prolonged with higher rates. Could the authors also comment on that issue?

We thank the reviewer for the comment. The ECG tracings were acquired on admission. Indeed, PR interval is rate dependent and this was stated in the manuscript (page 7 row 147-149).

-Four patients from TICM cohort received pacemakers - what kind of devices were chosen in those patients for pace and ablate strategy? Were they CRT devices or conventional pacemakers? Moreover, in Discussion the data on the % of patients with LBBB and CRT devices in DICM are provided, which are not listed in any Table/Results section

Patients in the pace and ablate strategy received different types of pacemakers. Three received Bi ventricular pacemakers, four received single chamber pacemaker and three received dual chamber pacemakers. We added device implantations to table supplemental 2 and to the result section in the text. As shown in supplemental 2, fourteen patients had CRTD implantation during follow up. Ten out of 14 patients had LBBB at presentation (page 23 row 509-510; page 8 row 179-181).

-What were the indications for implantation of an ICD in patients with TICM? If the criterion of >15% LVEF increase was fulfilled were they the subjects with initially so decreased LVEF that even after recovery of LV the LVEF remained <35%?

Thank you for your comment. Three TICMP patients underwent ICD implantation during follow up. One patient had very long QT with uncontrolled frequent premature ventricular beats and underwent dual chamber ICD implantation due to high risk of Torsades de Pointes (TdP).  The second patient underwent CRT-D implantation due to episodes of long short sequences leading to non-sustained TdP. The third patient underwent dual chamber ICD for secondary prevention after experiencing myocardial infarction and sustained ventricular tachycardia. We added this information to the manuscript (page 8 row 173-179).

-In Table 5 the data do not sum up to 100% in any of the columns. Could the authors explain this statistics? Was it that the sums should not include respectively 66 and 64, but only No of patients who experienced HF hospitalization?

We apologize for this misunderstanding. The numbers represent those patients with first heart failure recurrence following the index admission. In keeping with the defined end point, the percentages represent the absolute numbers from the total cohort.  

-No cause of death is presented, which in my opinion should be considered a limitation, especially when analyzing the outcome-oriented study.

We agree with the reviewer comment, and added this statement to the limitation section (page 12 row 298-300).

Reviewer 2 Report

jcm-2156661 “Long-term outcomes of tachycardia induced cardiomyopathy compared with idiopathic dilated cardiomyopathy” by Katz et al. aimed to compare the clinical presentation, comorbidities, and long-term outcomes of TICMP patients with those of IDCM patients.

1.       Title: Please remove the period (.) at the end of the manuscript title.

2.       Author Affiliations: Please assign asterisk (*) to the corresponding author.

3.       Introduction:

-          Page 2, lines 47-49: “The objective of our study was to assess …. with IDCM patients”. To avoid redundancy, please remove this sentence as it was already mentioned in the “Objective” section (Page 2, lines 50-51).

4.       Methods:

-          Study population: The reviewer suggests adding the N number of TICMP (n=64) and IDCM (n=66) patients.

-          Page 2, line 71: Please add comma between “During the index hospitalization” and “patients received”.

-          Statistical Analysis (Page 2, Lines 84-87): “Student’s t test or Mann–Whitney test for continuous parameters, as appropriate………..normally distributed continues variables were reported as mean ± 1 SD”. Do the authors indeed or performed student t test or report any data in the current manuscript as “mean ± SD”? if yes, please clarify which data set? If nothing reported, please edit statistical analysis to better reflect that.

-          Please replace “continues” by “continuous” and remove “1” in “mean ± 1 SD

5.       Tables and Figures:

-          All tables and figures including supplementary tables:

a-       Please spell out ALL abbreviations in alphabetical order, including those which are already spelled out in text (IDCM and TICMP).

b-      The reviewer suggests adding the corresponding statistical analysis in the footnote of all tables.

c-       Please indicate the significant p value by asterisk (*) or any equivalent symbol, when applicable.

-          Figure 1:

Pease edit “TIC” to “TICMP” for consistency.

-          Table 2:

a.       There is discrepancy between the values in Table 2 and their corresponding in text, in several places. The authors should detect the inaccurate values and correct them:

                                 I.            The median heart rate of IDCM group is listed as 82 BPM in the table, while mentioned as 87 BPM in the text (Page 5, line 128).

                               II.            The values of LVEDD and LVESD are different between table and text (page 6, lines 143-146).

                             III.            The p value of LBBB is listed as 0.003 in the table, while mentioned as 0.004 in the text (Page 6, line 149).

                            IV.            The median PR interval of IDCM is listed as 158 ms in the table, while mentioned as 159 ms in the text (Page 6, line 152).

b.       LBBB morphology and RBBB morphology, what are the numbers in the table refer to? Is it the number of patients? If yes, then please remove the word “morphology” as it gives indication that these numbers describe morphology.

c.       In general, since the values are already exhibited in the tables. The reviewer suggests just highlighting the significantly different parameters without repeating the numbers once again in the text.

-          Table 3:

a.       Please separate the title of Table 3 from the text.

b.       Please indicate the parameters by (%), wherever applicable.

c.       What is the statistical analysis used in this table? Please explain how p value is calculated for such low N number.

6.       Results:

-       The authors mentioned the hospitalization time was between January 2007 and June 2017 in the “Methods” (page 2, lines 56-57), while it was mentioned in the “Results” between March 2007 and June 2017 (Page 3, Line 110). Which one is more accurate? Please correct accordingly.

-       Page 6, lines 149-150: “The QRS duration was also significantly longer compared to TICMP patients (98 ms [86-111] vs 106 ms [96-148], p=0.002)”. Since the authors mentioned IDCM compared to TICMP, then the values should be listed respectively (106 ms vs 98 ms).

-       Page 6, Line 159: Please spell out the abbreviation (CKD).

-       Page 6, Line 167: “(35% [IQR 20-45%], 55% [IQR 47-60%], TICMP respectively, P<0.001). Please remove “TICMP”.

-       Page 6, lines 153-159: The authors mentioned the coronary artery disease assessment but did not list the findings of this assessment. Please clarify? Also, the authors explained why ischemic evaluation is missing in 2 patients from IDCM group but did not explain why the evaluation was not presented for 11 out of 64 patients of TICMP group. Please comment?

-       Page 6, lines 169-170: The change in LVEF% in TICMP group was from 30% [20-32%] to 55% [47-60%]. Could the authors confirm the median improvement of LVEF% for TICMP group was indeed, 25%?

-       Page 6, lines 170-173: The authors refer to 10 pacemakers as (16%). It would be better if the percentage represent off the TICMP patients implanted with pacemakers only (n=12) not the total number of TICMP patients. Also, please try to edit the sentence to reflect that there were total of 12 pacemakers were implanted in TICMP patients.

7.       Discussion:

-       The reviewer suggest discussing the findings of the current manuscript in relation to the results of these 2 relevant studies: DOI: 10.1002/clc.20161 (reference #12) and DOI: 10.1016/j.rec.2017.06.003 (Reference #28), wherever applicable.

-       Page 9, lines 214-218: “These studies, however, ……. resulting in inherent selection bias”. Please cite the proper reference(s)/studies.

-       Page 10, lines 254/ Table 2: While IDCM patients had showed wider QRS intervals (106 [138-174]) compared to those of TICMP patients (98 [86-111]), yet the upper limit for normal QRS duration has been defined to be less than 110 ms in adults (in other words; may be wider QRS but still within normal). The authors may consider reporting and compare how many patients in both groups had, indeed, prolonged/widened QRS complex (>110 ms).

-       Page 10, lines 258-260: “Patients who received bi-ventricular defibrillator had an improvement in their ejection fraction, however their LVEF remained reduced even after device implantation”. Do the authors mean that LVEF improved but not returned to normal/baseline? Please clarify.

-       Page 10, lines 261-263: “Moreover, it is strongly supporting their diagnosis of IDCM ….. after bi-ventricular defibrillator implantation”. The authors may need to rephrase/reassess this statement as the response to cardiac resynchronization therapy (CRT) may vary greatly between patients, where Lack of therapeutic response is regularly observed in about one-third of patients subjected to CRT (non-responders), although they are selected according to standard criteria relying on QRS duration.

8.       Supplementary Material

-          The reviewer suggests adding the manuscript title and authors names to the supplementary material.

-          Table S1: Please correct the table title.

-          Table S1 & Table S2: The abbreviations should be in alphabetical order. All abbreviations should be spelled out, including those which already spelled out in text (IDCM and TICMP).

-          Table S2: Please add [IQR] for the values, wherever applicable.

Author Response

Comments and Suggestions for Authors

jcm-2156661 “Long-term outcomes of tachycardia induced cardiomyopathy compared with idiopathic dilated cardiomyopathy” by Katz et al. aimed to compare the clinical presentation, comorbidities, and long-term outcomes of TICMP patients with those of IDCM patients.

  1. Title: Please remove the period (.) at the end of the manuscript title.

This was corrected according to the reviewer advise (page 1 row 1).

  1. Author Affiliations: Please assign asterisk (*) to the corresponding author.

This was corrected according to the reviewer advise (page 1 row 3).

  1. Introduction:

-          Page 2, lines 47-49: “The objective of our study was to assess …. with IDCM patients”. To avoid redundancy, please remove this sentence as it was already mentioned in the “Objective” section (Page 2, lines 50-51).

This was corrected according to the reviewer advise (page 3 row 52). 

  1. Methods:

-          Study population: The reviewer suggests adding the N number of TICMP (n=64) and IDCM (n=66) patients.

This was corrected according to the reviewer advise (page 4 row 70-71).

-          Page 2, line 71: Please add comma between “During the index hospitalization” and “patients received”.

This was corrected according to the reviewer advise (page 4 row 86).

-          Statistical Analysis (Page 2, Lines 84-87): “Student’s t test or Mann–Whitney test for continuous parameters, as appropriate………..normally distributed continues variables were reported as mean ± 1 SD”. Do the authors indeed or performed student t test or report any data in the current manuscript as “mean ± SD”? if yes, please clarify which data set? If nothing reported, please edit statistical analysis to better reflect that.

Thank you for your comment. All the parameters were tested for normality. However, it was decided to present the data as median and IQR for unity. This was corrected according to the reviewer advise (page 5 row 100-101).

-          Please replace “continues” by “continuous” and remove “1” in “mean ± 1 SD”

This was corrected according to the reviewer advise (page 5 row 100-101).  

  1. Tables and Figures:

-          All tables and figures including supplementary tables:

a-       Please spell out ALL abbreviations in alphabetical order, including those which are already spelled out in text (IDCM and TICMP).

We thank the reviewer for his comment. We changed the manuscript accordingly (page 16 row 408-409; page 17 row 413; page 18 row 419; page 19 row 424; page 20 row 432-435; page 21 row 447-454; page 22 row 465-467; page 22 row 473; page 23 row 489-490; page 24 row 513-514; page 24 row 519-524).

b-      The reviewer suggests adding the corresponding statistical analysis in the footnote of all tables.

We thank the reviewer for his comment. We changed the manuscript accordingly (page 20 row 436-437; page 21 row 455-456; page 22 row 468; page 22 row 474; page 24 row 525-526).  

c-       Please indicate the significant p value by asterisk (*) or any equivalent symbol, when applicable.

We thank the reviewer for his comment. We changed the manuscript accordingly (page 20 row 431; page 21 row 446; page 22 row 472; page 24 row 518).

-          Figure 1:

Pease edit “TIC” to “TICMP” for consistency.

We thank the reviewer for his comment. We changed the manuscript accordingly (page 16 row 406).

-          Table 2:

  1. There is discrepancy between the values in Table 2 and their corresponding in text, in several places. The authors should detect the inaccurate values and correct them:

  1.     The median heart rate of IDCM group is listed as 82 BPM in the table, while mentioned as 87 BPM in the text (Page 5, line 128).

We thank the reviewer for his comment. We changed the manuscript accordingly (page 6 row 134).

  1.             The values of LVEDD and LVESD are different between table and text (page 6, lines 143-146).

We thank the reviewer for his comment. We changed the manuscript accordingly (page 6 row 140-142).  

                             III.            The p value of LBBB is listed as 0.003 in the table, while mentioned as 0.004 in the text (Page 6, line 149).

We thank the reviewer for his comment. We changed the manuscript accordingly (page 6 row 146).

  1. The median PR interval of IDCM is listed as 158 ms in the table, while mentioned as 159 ms in the text (Page 6, line 152).

We thank the reviewer for his comment. We changed the manuscript accordingly (page 7 row 148).

  1. LBBB morphology and RBBB morphology, what are the numbers in the table refer to? Is it the number of patients? If yes, then please remove the word “morphology” as it gives indication that these numbers describe morphology.

The number represent patients' numbers, and we changed the manuscript according to the reviewer suggestion (page 21 row 446).

  1. In general, since the values are already exhibited in the tables. The reviewer suggests just highlighting the significantly different parameters without repeating the numbers once again in the text.

We thank the reviewer for his comment. The data in the tables and figure is very comprehensive. We highlighted in the text significantly different parameters to allow the reader continuous reading.

-          Table 3:

  1. Please separate the title of Table 3 from the text.

We thank the reviewer for his comment. We changed the manuscript accordingly (page 22 row 463).

  1. Please indicate the parameters by (%), wherever applicable.

We thank the reviewer for his comment. We changed the manuscript accordingly (page 22 row 464).

  1. What is the statistical analysis used in this table? Please explain how p value is calculated for such low N number.

We are aware for the low N number in each group. We used Fisher’s exact test for the analysis (page 22 row 468).

  1. Results:

-       The authors mentioned the hospitalization time was between January 2007 and June 2017 in the “Methods” (page 2, lines 56-57), while it was mentioned in the “Results” between March 2007 and June 2017 (Page 3, Line 110). Which one is more accurate? Please correct accordingly.

We apologize for this mistake. The time frame is between March 2007 and June 2017. We corrected the manuscript accordingly (page 4 row 72).

-       Page 6, lines 149-150: “The QRS duration was also significantly longer compared to TICMP patients (98 ms [86-111] vs 106 ms [96-148], p=0.002)”. Since the authors mentioned IDCM compared to TICMP, then the values should be listed respectively (106 ms vs 98 ms).

We thank the reviewer for his comment. We changed the manuscript accordingly (page 7 row 147).

-       Page 6, Line 159: Please spell out the abbreviation (CKD).

We thank the reviewer for his comment. We changed the manuscript accordingly (page 7 row 155).

-       Page 6, Line 167: “(35% [IQR 20-45%], 55% [IQR 47-60%], TICMP respectively, P<0.001). Please remove “TICMP”.

We thank the reviewer for his comment. We changed the manuscript accordingly (page 7 row 165).

-       Page 6, lines 153-159: The authors mentioned the coronary artery disease assessment but did not list the findings of this assessment. Please clarify? Also, the authors explained why ischemic evaluation is missing in 2 patients from IDCM group but did not explain why the evaluation was not presented for 11 out of 64 patients of TICMP group. Please comment?

Coronary artery disease assessment was performed to the vast majority of patients with IDCM as mentioned in the text. Classification as tachycardia induced cardiomyopathy does not require coronary artery exclusion unless there is high suspicion that ischemia is resulting in cardiomyopathy or lead to lack of improvement in LVEF after adequate rate or rhythm control. Those 11 patients normalized their LVEF during follow up and hence coronary artery disease at that point was not necessary to confirm this etiology.  

-       Page 6, lines 169-170: The change in LVEF% in TICMP group was from 30% [20-32%] to 55% [47-60%]. Could the authors confirm the median improvement of LVEF% for TICMP group was indeed, 25%?

Yes, during follow-up the median LVEF was 55 %[47-60%] in TICMP patients and 35% [20-45%] in IDCM patients. The absolute median improvement in TICMP group was 25% [18-30%].

-       Page 6, lines 170-173: The authors refer to 10 pacemakers as (16%). It would be better if the percentage represent off the TICMP patients implanted with pacemakers only (n=12) not the total number of TICMP patients. Also, please try to edit the sentence to reflect that there were total of 12 pacemakers were implanted in TICMP patients.

We thank the reviewer for his comment. We changed the manuscript accordingly (page 7 row 168-170).

  1. Discussion:

-       The reviewer suggest discussing the findings of the current manuscript in relation to the results of these 2 relevant studies: DOI: 10.1002/clc.20161 (reference #12) and DOI: 10.1016/j.rec.2017.06.003 (Reference #28), wherever applicable.

We thank the reviewer for his comment. We included reference #12 in our discussion (page 11 row 255-257).

-       Page 9, lines 214-218: “These studies, however, ……. resulting in inherent selection bias”. Please cite the proper reference(s)/studies.

We thank the reviewer for his comment. We changed the manuscript accordingly (page 9 row 211).  

-       Page 10, lines 254/ Table 2: While IDCM patients had showed wider QRS intervals (106 [138-174]) compared to those of TICMP patients (98 [86-111]), yet the upper limit for normal QRS duration has been defined to be less than 110 ms in adults (in other words; may be wider QRS but still within normal). The authors may consider reporting and compare how many patients in both groups had, indeed, prolonged/widened QRS complex (>110 ms).

We thank the reviewer for this comment. Twenty-six IDCM vs 18 TICMP patients had QRS equal or greater than 110 ms. However, the differences were not statistically significant.  

-       Page 10, lines 258-260: “Patients who received bi-ventricular defibrillator had an improvement in their ejection fraction, however their LVEF remained reduced even after device implantation”. Do the authors mean that LVEF improved but not returned to normal/baseline? Please clarify.

Patients with LBBB and DCM who were implanted with bi-ventricular defibrillator had minor improvement in their LVEF. This observation is in keeping with the diagnosis of IDCM versus LBBB induced cardiomyopathy, where in the latter, LVEF is significantly improved. The statement was rephrased in the text (page 10-11 row 248-252).

-       Page 10, lines 261-263: “Moreover, it is strongly supporting their diagnosis of IDCM ….. after bi-ventricular defibrillator implantation”. The authors may need to rephrase/reassess this statement as the response to cardiac resynchronization therapy (CRT) may vary greatly between patients, where Lack of therapeutic response is regularly observed in about one-third of patients subjected to CRT (non-responders), although they are selected according to standard criteria relying on QRS duration.

We thank the reviewer for his comment. We rephrased the statement in the text. The purpose of the statement was to repel possible claim that some of the patients with DCM and LBBB are not true IDCM but LBBB mediated cardiomyopathy.

We agree with the reviewer statement that one third of CRT recipients are non-responders. This, however, it is very uncommon in LBBB cardiomyopathy patients. They usually show much more improvement in LVEF.  

  1. Supplementary Material

-          The reviewer suggests adding the manuscript title and authors names to the supplementary material.

We thank the reviewer for his comment. We changed the manuscript accordingly (page 23 row 498-503).

-          Table S1: Please correct the table title.

We thank the reviewer for his comment. We changed the manuscript accordingly (page 24 row 511).

-          Table S1 & Table S2: The abbreviations should be in alphabetical order. All abbreviations should be spelled out, including those which already spelled out in text (IDCM and TICMP).

We thank the reviewer for his comment. We changed the manuscript accordingly (page 16 row 408-409; page 17 row 413; page 18 row 419; page 19 row 424; page 20 row 432-435; page 21 row 447-454; page 22 row 465-467; page 22 row 473; page 23 row 489-490; page 24 row 513-514; page 24 row 519-524).

Round 2

Reviewer 2 Report

The reviewer would like to thank the authors for implementing the suggestions/corrections per first round of review. Below few minor comments/suggestions that might be further beneficial to the manuscript:

1)    Tables:

a. Please indicate for continuous variables by “median [IQR]”, whereas categorical variables by “n (%)”. Example: “Sex, Male; n (%)” and “Age, Years; median [IQR]”. That would apply to ALL tables including supplementary tables, wherever applicable.

b. Table 1: The reviewer suggests moving “BMI” to the top of the table below “age”.

c. Table 5: The authors mentioned in the footnote the number of patients who had HF recurrence during follow up. The reviewer also suggests adding a new row (first row) in the table reflecting that as well.

d. Supplementary Table 1: Please make sure the font style and size are consistent throughout the table.

e. Page 3, line 118, Page 6, line 171, and page 10, line 271: In matter of referring to supplementary tables, it is better to use “Table S1” and “Table S2” instead of “Supplemental 1” and “Supplemental 2”, respectively, unless journal guidelines state otherwise.

2) Statistical analysis:

a. Page 2, line 81: Reviewer suggests editing “Non-normal distributed” to “The non-normally distributed”.

b. Page 2, line 83: Please remove additional “a” letter after [Q1-Q3].

3) Abbreviations:

a. Versus and vs: Please use only one format consistently throughout the manuscript.

b. Page 6, line 187: “LBBB”: upon addition a new paragraph to “Follow-up”, this is the first appearance of “LBBB” abbreviation in text. Please spell it out accordingly.

c. CRTP and CRTD: Could the authors clarify whether C letter in both abbreviations refers to “continuous” or “cardiac”? Please edit if needed.

4) Results/Discussion:

The authors reported in their rebuttal: “Twenty-six IDCM vs 18 TICMP patients had QRS equal or greater than 110 ms. However, the differences were not statistically significant”. The reviewer strongly suggests adding this finding to the manuscript.

Author Response

Title

Long-term outcomes of tachycardia induced cardiomyopathy compared with idiopathic dilated cardiomyopathy.

Authors

Moshe Katz*, Amit Meitus , Michael Arad , Anthony Aizer , Eyal Nof , Roy Beinart

Reviewer#2

The reviewer would like to thank the authors for implementing the suggestions/corrections per first round of review. Below few minor comments/suggestions that might be further beneficial to the manuscript:

1)    Tables:

  1. Please indicate for continuous variables by “median [IQR]”, whereas categorical variables by “n (%)”. Example: “Sex, Male; n (%)” and “Age, Years; median [IQR]”. That would apply to ALLtables including supplementary tables, wherever applicable.
  2. Table 1: The reviewer suggests moving “BMI” to the top of the table below “age”.
  3. Table 5: The authors mentioned in the footnote the number of patients who had HF recurrence during follow up. The reviewer also suggests adding a new row (first row) in the table reflecting that as well.
  4. Supplementary Table 1: Please make sure the font style and size are consistent throughout the table.
  5. Page 3, line 118, Page 6, line 171, and page 10, line 271: In matter of referring to supplementary tables, it is better to use “Table S1” and “Table S2” instead of “Supplemental 1” and “Supplemental 2”, respectively, unless journal guidelines state otherwise.

We thank the reviewer for his comments. We changed the text and table according to the reviewer suggestions:

  1. Page 3 line 113, page 5 line 133, page 7 lines 209 and 216 and page 1 on supplemental material
  2. Page 3 line 113
  3. Page 7 line 216
  4. Page 1 on supplemental
  5. Page 3 line 112, page 6 line 173, page 10 line 273

2) Statistical analysis:

  1. Page 2, line 81: Reviewer suggests editing “Non-normal distributed” to “Thenon-normally distributed”.
  2. Page 2, line 83: Please remove additional “a” letter after [Q1-Q3].

We corrected the text according to the reviewer's suggestions.

3) Abbreviations:

  1. Versus and vs: Please use only one format consistently throughout the manuscript.
  2. Page 6, line 187: “LBBB”: upon addition a new paragraph to “Follow-up”, this is the first appearance of “LBBB” abbreviation in text. Please spell it out accordingly.
  3. CRTP and CRTD: Could the authors clarify whether C letter in both abbreviations refers to “continuous” or “cardiac”? Please edit if needed.

We thank the reviewer for his important comments. We corrected the text according to the reviewer's comments.

  1. Page 3 line 109, page 4 line 130, page 5 lines 131-132, page 5 lines 146-149 151-152, page 6 lines 156 and 199
  2. Page 6 190
  3. Page 6 lines 168 and 185

4) Results/Discussion:

The authors reported in their rebuttal: “Twenty-six IDCM vs 18 TICMP patients had QRS equal or greater than 110 ms. However, the differences were not statistically significant”. The reviewer strongly suggests adding this finding to the manuscript.

We added this finding to the text.

Page 6 153-155